# DIFFERENTIALLY PRIVATE MODEL COMPRESSION VIA SELECTIVE PRETRAINING

## ABSTRACT

Suppose we want to train text prediction models in email clients or word processors. These models, which serve billions of predictions per hour, must preserve the privacy of user data and adhere to specific model size constraints to meet memory, inference time requirements, and to reduce inference cost. Building small, fast, and private domain-specific language models is a thriving area of research. In this work, we show that a careful pre-training on a *subset* of the public dataset that is guided by the private dataset is crucial to train small DP language models. On standard benchmarks, models trained with our new framework achieve state-of-the-art performance, improving upon all the baselines from the literature.

Besides performance improvements, our framework also shows that with careful pre-training and private fine-tuning, smaller models can match the performance of much larger models that do not have access to private data, highlighting the promise of private learning as a tool for model compression and efficiency.

## 1 INTRODUCTION

Many papers have shown that deep learning models are vulnerable to attacks aimed at extracting information from the training data (Shokri et al., 2017; Hayes et al., 2019; Carlini et al., 2021; Zhang et al., 2021; Choquette-Choo et al., 2021; Carlini et al., 2023; Matsumoto et al., 2023). A provable path for mitigating such privacy attacks is to train the models with *differential privacy* (DP) Dwork et al. (2006), a mathematically rigorous notion for quantifying the privacy leakage of a machine learning model. Over the past few years, there has been a rapid progress in our understanding of deep learning with DP, both in terms of computational efficiency (He et al., 2023; Bu et al., 2021; Lee and Kifer, 2021; Subramani et al., 2021; Anil et al., 2022) and privacy-utility trade-off (De et al., 2022; Zhou et al., 2021; Zhu et al., 2020; Golatkar et al., 2022; Sander et al., 2022; Bu et al., 2022a; Panda et al., 2022; Luo et al., 2021; Kairouz et al., 2021). One of the findings of these works has been that pre-training (or pre-trained models) is crucial for maximizing performance. There is some theoretical evidence of why and how pre-training helps private learning Li et al. (2022b); Ganesh et al. (2023).

Most of the DP literature mentioned above focus on settings where *inference time* is not a bottleneck and one can deploy models of *any size*. In such a case, existing evidence is that larger models pre-trained on vast amounts of public data perform better when combined with private fine-tuning (Li et al., 2022c; Yu et al., 2022; Mehta et al., 2022). However, there are plenty of applications where the size of the model is restricted by the *inference time;* think of a language model of an email client or a face identification model running in a security system. In such applications, if the inference time is not good then the quality of predictions becomes irrelevant. Further, note also that in both these applications the training data is quite sensitive, and the models should protect the privacy of users. Building small, fast, and private domain specific language models is also a thriving area in industry with several start-ups (MosiacML; ScaleAI). There is also economic motivation as smaller models offer cheaper inference costs.

Recently, Mireshghallah et al. (2022) explored this question by considering model compression strategies for during the *private fine-tuning* stage. They studied several DP variants of standard algorithms used for model compression from the non-private world, and reached two broad conclusions. First, knowledge distillation algorithm (Hinton et al., 2015), arguably the most widely used algorithm for model compression, is not as effective when applied within the context of private learning. Using their best variant of private knowledge distillation algorithm, 50% compression of

a BERT-base model leads to a 6.4% relative performance drop on the MNLI dataset, which is only slightly better than directly fine-tune the smaller model with DP. Second, they showed that DP pruning algorithms, compared to knowledge distillation, tend to perform better for model compression; however, achieving *structured sparsity*, which is crucial for inference time, remains a challenge. These observations motivate an intriguing question: *is there a fundamental roadblock to training efficient and high-performing DP language models?*

## 1.1 OUR CONTRIBUTIONS

The main technical contribution of our work is to bring the focus on pre-training (on the public data) towards building smaller DP language models. To our knowledge, this is the *first* work in DP literature that explores better pre-training of LLMs guided by private data. Taking cues from neural scaling laws and data pruning (Sorscher et al., 2022; Hoffmann et al., 2022), we observe that the number of pre-training tokens that achieves the optimal pre-training loss is a function of model size. This can be (loosely) inferred from Figure 1, where for a 21 million parameter model, pre-training on random 15% tokens on the public dataset is no different than pre-training on the full dataset, and more clearly in Figure 5. Intuitively, smaller models have smaller capacity, and hence pre-training on all the available public data does not always lead the best downstream task performance. Then, how to identify the *best subset of tokens for pre-training*? Very recent work in non-private literature has tackled this question using various data filtering approaches (Sorscher et al., 2022; Abbas et al., 2023; Xie et al., 2023). Our hypothesis is that when there is access to high-quality private data, one can use the private data to guide pre-training data selection.

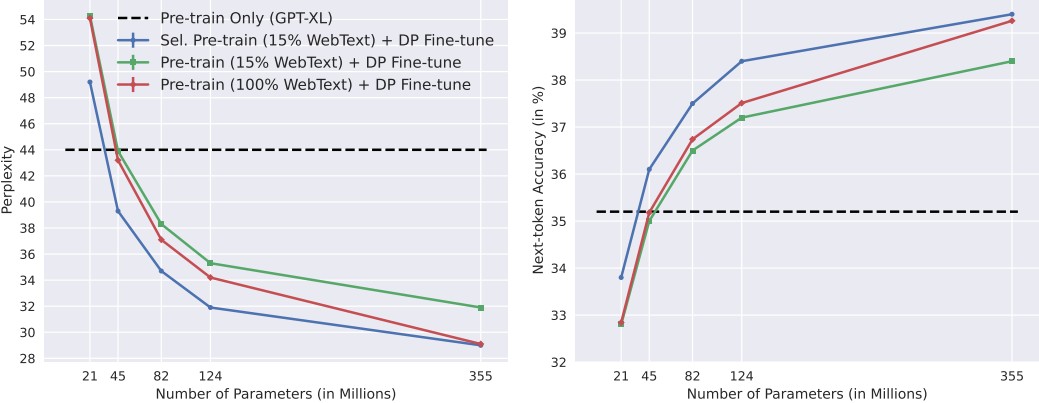

Figure 1: A representative result from our findings. We plot perplexity and top-1 next word accuracy of GPT models on the test set of the Enron email dataset. Overall privacy budget is $(\varepsilon = 7.3, \delta = 1 \times 10^{-7})$. The dashed line shows the zero-shot performance of GPT2-XL with 1.5 billion parameters. The figure shows that our framework yields clear improvements in both perplexity and next-token prediction accuracy, which can significantly improve the overall model behavior.

**A Framework for Private Training To Maximize Downstream Performance** We give a new framework for training domain-specific language models of a given size with differential privacy. In our framework, a privacy-preserving algorithm is used to select *a subset* of public data for pre-training, which we call *selective pre-training*. This step is followed by private fine-tuning via DP-SGD. Our experiments demonstrate that selective pre-training is important for smaller models to achieve better performance when privately fine-tuned with DP-SGD. Figure 1 is a representative summary of our findings. The main takeaway is the following:

*Our new framework leads to clear improvements in the downstream task performance on standard NLP benchmarks, both in perplexity and prediction accuracy, for all model sizes we experimented.*

**State-of-the-art for Small Models** We do an empirical evaluation of our framework and show that our method is superior to baselines known in the literature. We achieve state-of-the-art results on standard benchmarks, with significant improvements compared to the previous best results in Mireshghallah et al. (2022).

**Private Algorithms for Dataset Selection** There is a growing interest in understanding the right data selection for pre-training both for general domain models (GPTs, PaLM, LLaMA) and specific models (Khanmigo, Codex, AlphaFold) in non-private deep learning literature; see Xie et al. (2023); Gururangan et al. (2020); Jain et al. (2023) and references therein. It is reasonably well established that the pre-training dataset has a significant impact on the downstream performance of the model and its transfer learning abilities. However, unlike these works, our focus is on right data selection for pre-training as a function of model size. As a byproduct of our framework, we introduce these problems to private learning literature, show a simple recipe for data selection that is more friendly for private learning. Our experiments in Section 4.1 and 4.2 bring to light the following phenomenon:

*The benefits of selective pre-training are greater for private deep learning compared to non-private deep learning.*

**Differential Privacy as a Tool For Model Compression** Consider Figure 1 again. Observe that the performance of a small model with only 45 million parameters is comparable to the zero-shot performance of GPT2-XL public model with 1.5 billion parameters. Furthermore, smaller DP models with 82 and 124 million parameters that were trained using our new framework match or surpass the perplexity and top-1 accuracy of *larger DP models* with 124 and 355 million parameters respectively that were not carefully pre-trained. These observations have remarkable consequences for training domain-specific models in real-world applications.

*In all the scenarios where differential privacy provides meaningful protection of data privacy, a small amount of private data along with our selective pre-training followed by private fine-tuning can lead to a substantial compression of the model sizes, thus improving the inference time and reduction in the inference cost.*

In other words, intuitively speaking, a model that has no access to high-quality data has to be larger to generalize, whereas smaller models that have access to high-quality data can outperform bigger models in domain-specific tasks. Thus, differentially private training (combined with our framework) can truly unlock the value of high-quality but sensitive data. We anticipate that this conceptual message of private learning as a tool for model efficiency, which to our knowledge has not been emphasized before in the literature, will find more applications in deep learning. On a scientific front, it brings the data quality aspect to focus in understanding deep learning as observed in some other recent works too Abbas et al. (2023); Sorscher et al. (2022).

**Real-world Impact** Our framework was recently used in training an industry grade differentially private text prediction language model that now serves many NLP applications in a big AI company. As text prediction models (on email clients/servers, word processors, etc.) serve billions of queries per hour, the inference cost savings due to the decrease in model size are significant. Further, due to better inference time, online performance metrics, such as the number of predictions accepted by the users, also improve. While we do not think these facts as a scientific contribution of our work, it highlights the efficacy of our framework for real-world datasets and applications, besides the standard benchmarks considered in this paper.

Compared to prior work that can utilize off-the-shelf pre-trained models (Mireshghallah et al., 2022), our methods pre-train models from scratch, thereby incurring an additional computational cost associated with pre-training. However, in many real world applications, the cost of training (pre-training, fine-tuning, data selection, etc.) is a negligible fraction of the inference cost. The training cost is an one time investment, whereas the inference cost accumulates over the time. Finally, our work is at the intersection of model compression, data-selection for pre-training, and domain adaptation literature. We discuss how our work is different from the previous works in Section 5.

## 1.2 IMPACT OF LOW QUALITY PRE-TRAINING DATA

Our work shows that better pre-training data prepares language models for better fine-tuning with DP-SGD. In the absence of high quality pre-training data, however, to see the benefits of pretraining for private learning, *one needs significantly larger models trained on vast amounts of public data*. This is consistent with what previous papers have shown (Li et al., 2022c; Yu et al., 2022; Mehta et al., 2022; De et al., 2022). We demonstrate these intuitions by performing the following experiment.

**Spanish GPT for English:** We start with a 739M GPT model that is pre-trained from scratch on Spanish corpus (Gutiérrez-Fandiño et al., 2021), which acts as low quality pre-training data. Then we

fine-tune it on the Enron email dataset, which is an English-language corpus. The privacy parameters are the same as those in Figure 1. The model achieves 37.7% accuracy on next-word prediction, which is comparable to the performance of an 124M GPT model that uses standard pre-training on English corpus, as well as the performance of a 82M GPT model trained with selective pre-training. Another interesting thing to note is that pre-training still offers benefits, as the accuracy of a randomly initialized 739M GPT model when privately fine-tuned on the Enron dataset only achieves an accuracy of only 17.2%.

## 1.3 PRELIMINARIES

We begin with the formal definition of differential privacy.

**Definition 1** ( $(\varepsilon, \delta)$-Differential Privacy (DP) (Dwork et al., 2006)). *A randomized algorithm $\mathcal{A}$ is $(\varepsilon,\delta)$-differentially private if for any two neighboring datasets $D$ and $D'$, which differ in exactly one datapoint, and for every subset $\mathcal{S}$ of possible outputs:* $\Pr[\mathcal{A}(D) \in \mathcal{S}] \leq e^{\varepsilon} \Pr[\mathcal{A}(D') \in \mathcal{S}] + \delta$.

**Private Deep Learning:** In the context of deep learning, DP guarantees that the trained model *weights* are private with respect to a training dataset, and hence can be released publicly. To train a deep learning model with privacy, the most popular method is to first release the gradients of an optimizer with differential privacy and then update the model with privatized gradients (Song et al., 2013; Bassily et al., 2014; Abadi et al., 2016). We follow the approach in Abadi et al. (2016) to make gradients differentially private. Abadi et al. (2016) augment each minibatch of gradients with per-example gradient clipping and Gaussian noise addition steps. The clipping step ensures that no one user's sample significantly changes the weights of the model and the noise added guarantees that the contribution of a single example is masked.

## 2 PROBLEM STATEMENT AND OUR ALGORITHMIC FRAMEWORK

Input to our problem is a private dataset $D_{\text{priv}}$ corresponding to a downstream task $T$, a model $M$ of size $p$, privacy parameters $\varepsilon > 0$, $\delta > 0$, and a public dataset $D_{\text{pub}}$. Our goal is to train $M$ on public and private datasets with the aim of maximizing the downstream performance on the task $T$. The entire process should be $(\varepsilon, \delta)$-differentially private with respect to $D_{\text{priv}}$. The constraint on model size is important to compare various algorithms in our setting. In applications, the constraints on model size arise naturally as a consequence of memory and/or inference time requirements.

Our framework for solving the problem consists of the following 3 steps.

1. ***Privacy Preserving Data Selection***: Given $D_{\text{priv}}$, invoke a privacy preserving data selection algorithm $\mathcal{A}_{\text{select}}$ to find a $D'_{\text{pub}} \subseteq D_{\text{pub}}$. The privacy budget for this step is $(\varepsilon_1, \delta_1)$.

2. ***Non-Private Pre-training***: Pre-train the model $M$ on $D'_{\text{pub}}$ with a standard pre-training algorithm. This step does not consume any privacy budget.

3. ***Private Fine-tuning***: Fine-tune $M$ on $D_{\text{priv}}$ with a differentially private algorithm $\mathcal{A}_{\text{finetune}}$. The privacy budget for this step is $(\varepsilon_2, \delta_2)$.

The non-private pre-training step can be viewed as a post-processing function to $\mathcal{A}_{\text{select}}$ and thus no privacy budget is consumed. The advanced composition theorem of DP (see (Steinke, 2022) for example) guarantees that our framework is $(\varepsilon, \delta)$-DP. In our experiments, we use the Privacy Random Variable (PRV) Accountant (Gopi et al., 2021; Ghazi et al., 2022; Koskela et al., 2020). The PRV accountant gives tighter bounds on privacy parameters $\varepsilon$ and $\delta$ than the moments accountant in Abadi et al. (2016). The rest of the paper is devoted to describing the first step of our framework, followed by experiments to verify the effectiveness of our methods on different datasets.

## 3 PRIVACY PRESERVING DATA SELECTION

We describe our approach to implementing a privacy-preserving data selection algorithm. We provide a specific implementation of our framework and demonstrate its effectiveness, however, our approach is general and can be combined with other private data selection algorithms.

### 3.1 OUR IMPLEMENTATION OF DATA SELECTION

Our framework is loosely inspired by the data cleaning framework used in GPT3 and PaLM models Brown et al. (2020); Chowdhery et al. (2022), although motivations are a bit different. The classifiers in Brown et al. (2020); Chowdhery et al. (2022) are trained to filter out noisy documents from datasets. In fact, the source datasets in our paper, i.e., OpenWebText and Wikipedia, are considered positive examples in Brown et al. (2020). Our classifier is trained to recognize examples that are similar to samples in the target data. We initialize the classifier with a pre-trained LM and fine-tune it with differential privacy to predict whether a sentence is sampled from the distribution of the target data. We use the classifier to predict all sentences in the source data and *rank* them according to confidence scores. Although deep neural networks could be overconfident and need calibration in some applications (Guo et al., 2017; Zhang et al., 2022), not calibrating the outputs does not affect our algorithm because calibration does not change the relative ranking among sentences. We select the top sentences until we reach the target number of pre-training tokens. Figure 2 shows an overview of our implementation. In the non-private world, concurrent work of Xie et al. (2023) studies this problem and proposes an $n$-gram based approach; we believe that our technique is more suitable for DP world.

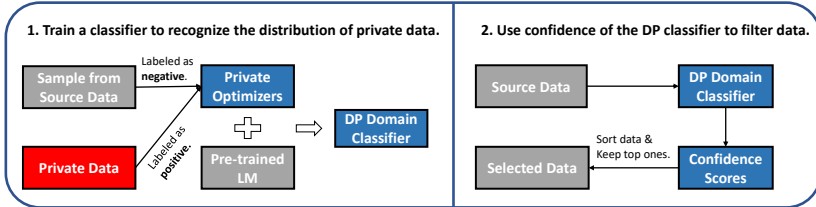

Figure 2: An illustration of the data selection process. We train a classifier to identify target examples and rank source examples based on the confidence scores of the classifier.

We create a training set to teach the classifier to recognize a target data distribution. Sentences in the target dataset are labelled as positive. Random samples from the source data are labelled as negative. It has been widely observed that a larger training set helps private learning (Bassily et al., 2014; Tramèr and Boneh, 2021). Therefore we set the number of negative examples as five times larger than the number of positive examples. The privacy cost of training the classifier is accounted in the overall privacy cost.

We run experiments with the Enron Email dataset[1] as the target and the OpenWebText dataset (Gokaslan and Cohen, 2019) as the source. The classifier is initialized with a 124M GPT model pre-trained on OpenWebText. With a single Tesla A100 GPU, it takes approximately one hour for training the classifier. With eight Tesla V100 GPUs, it takes less than two hours for computing the confidence scores for all sequences in OpenWebText. The privacy guarantee is $(0.7, 1 \times 10^{-8})$-DP if we only consider the privacy cost of this step. More implementation details are in Section 4.1. The trained classifier achieves an F1-score of 98.5%. The classifier achieves an F1-score of 92.7% if it is not initialized with a pre-trained LM.

We use the trained classifier to select 10% of OpenWebText. We plot the word clouds of Enron email, OpenWebText, and the selected subset of OpenWebText (Figure 3), to visually illustrate the dataset selected by our algorithm. The word clouds only show the nouns to exclude common prepositions and verbs. There are 28 nouns which appear in both the top 100 nouns of the Enron email dataset and the top 100 nouns of OpenWebText. The number of overlaps increases to 39 when comparing Enron email with the selected subset of OpenWebText, suggesting the trained domain classifier is an effective tool for data selection. In Appendix B.1, we also present the results of using GLUE (Wang et al., 2018) tasks as the targets and the pre-training corpus of BERT (Devlin et al., 2019) as the source.

---

[1] https://www.cs.cmu.edu/~enron/

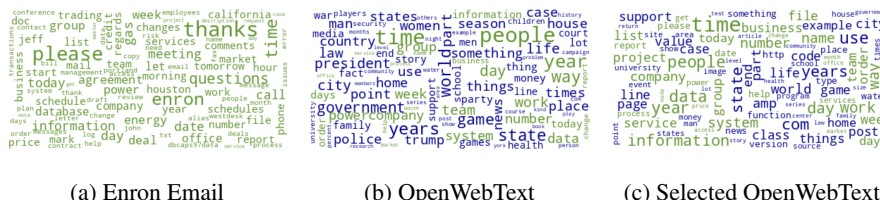

|                    |                    |                          |
| ------------------ | ------------------ | ------------------------ |
| (a) Enron Email    | (b) OpenWebText    | (c) Selected OpenWebText |

Figure 3: The 100 most frequent nouns in Enron email, OpenWebText, or a selected subset of OpenWebText (10%). A larger font size indicates that the word appears more frequently. Green words are the 100 most frequent nouns in Enron Email. OpenWebText and selected OpenWebText have 28 and 39 words, respectively, that are among the 100 most frequent nouns in Enron Email.

## 4 EXPERIMENTAL EVALUATION

We evaluate our full framework (Section 2) on language generation and understanding tasks, comparing on datasets that most previous works used (Li et al., 2022c; Yu et al., 2022). The goal here is to empirically verify the main claim made in the introduction: our framework can be used as an effective tool for model compression, and beats the existing baselines in the literature (Mireshghallah et al., 2022). We note that Mireshghallah et al. (2022) did experiments on GLUE benchmark only, and we compare against them in the next section. We begin with the language modeling on the email dataset, which was the motivating example from the real world application.

### 4.1 IMPLEMENTING THE FRAMEWORK ON THE ENRON EMAIL DATASET

Our first target task is causal language modeling on the Enron email dataset. The dataset contains approximately 0.5 million (M) emails written by employees of the Enron Corporation and is publicly available for research use. We choose this dataset because its distribution closely resembles some private datasets in the real world for which it is hard to find off-the-shelf pre-training data.

#### 4.1.1 EXPERIMENT SETUP

We briefly describe important parameters of our experimental setup. More details are in Appendix C.

**Target and Source Data** We divide the text into sequences of length 256 and let each sequence be a datapoint, which constitutes the granularity of our privacy guarantees. Although most of the emails in the Enron email datasets are shorter than hundred words, some extreme long emails may be splitted into multiple training sequences. In real world applications, it is important to carefully bound the maximum contribution of a single email/user. There are ∼70K sequences in total. We use 80% of them for training and evenly split the rest 20% for validation and testing. The source data is OpenWebText (Gokaslan and Cohen, 2019) which contains ∼4 billion tokens. The sequence size for OpenWebText is 512, following the choice in Radford et al. (2018).

**Models** Models in this section are from the GPT family (Radford et al., 2019). We change the number of layers, hidden size, and intermediate size of the fully connected block to get five different model sizes (21M, 45M, 82M, 124M, and 355M). Details of the models and pre-training hyperparameters are in Appendix C. All models are pre-trained with nodes with 8x Nvidia Tesla V100 GPUs.

**Data Selection** We use the algorithm in Section 3 to select 2M sequences from the source data for pre-training. We train the domain classifier for 3 epochs. The baselines include 1) pre-training with 2M random sequences and 2) pre-training with all of OpenWebText.

**Privacy Budget and Hyperparameters** The overall privacy budget is $(7.3, 1 \times 10^{-7})$-DP, similar to previous works on this topic (Li et al., 2022c; Yu et al., 2022). To reduce the privacy cost of hyperparameter tuning (Liu and Talwar, 2019; Papernot and Steinke, 2022; Mohapatra et al., 2022), we follow the findings in previous work to set most of the hyperparameters and only tune the learning rate to adapt to models of different sizes. The hyperparameters for private learning are listed in Table 4 in Appendix C.

### 4.1.2 SELECTIVE PRE-TRAINING PROVIDES CLEAR GAINS, MODEL EFFICIENCY

Figure 1 (see Section 1.1) shows the perplexity and next-word prediction accuracy of different models on the test split of the Enron email dataset. We also present the next-word accuracy and its standard deviation across random seeds in Table 1 in Appendix B.2 as a complementary to Figure 1. It is clear from the figure that our framework improves performance compared to existing techniques.

More significantly, we see that smaller models can match the performance of much larger models; for example, the 82M model using selective pre-training matches the 124M model using normal pre-training. *This shows that the proposed framework can be used to improve the efficiency-utility trade-off of private learning.* We also include the zero-shot performance of the off-the-shelf GPT2-XL model (1.5 billion parameters) in Figure 1. The zero-shot performance of GPT2-XL is worse than the models that have access to private data and are of much smaller size. These findings highlight the importance of private data, which can be loosely treated as high quality data, as well as the importance of privacy-enhancing technologies that facilitate the trustworthy use of such data. Figure 9 in Appendix B.2 also presents the results under different privacy budgets ($\varepsilon$ ranging from 2.3 to 10.9). We observe consistent gains when the selective pre-training framework is used.

### 4.1.3 SELECTIVE PRE-TRAINING IS MORE IMPORTANT FOR PRIVATE LEARNING COMPARED TO NON-PRIVATE LEARNING

We also fine-tune the models without differential privacy to see whether selective pre-training improves downstream performance in non-private learning. The results are in Figure 4. When using 15% of OpenWebText, selective pre-training still improves the performance of all models though the improvement is smaller compared to the private world. When using 100% of OpenWebText, the benefits of selective pre-training gradually diminish as the model size increases. This suggests that selective pre-training is more important for private learning. Our hyperparameters for non-private fine-tuning can be found in Appendix C.

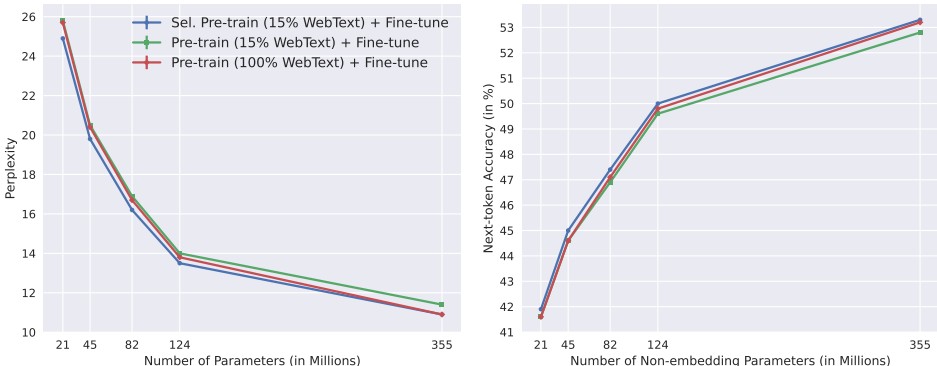

Figure 4: Perplexity and top-1 next word accuracy of GPT models on the test set of the Enron email dataset. The models are trained without DP. Selective pre-training still improves over standard pre-training, however, the improvements are smaller compared to private learning.

### 4.2 EXPERIMENTS ON GLUE

We conduct experiments on the GLUE benchmark (Wang et al., 2018), a common benchmark for fine-tuning language models with DP (Yu et al., 2021; Li et al., 2022c; Bu et al., 2022b). Our results show that selective pre-training also improves DP fine-tuning for language understanding tasks, beating the baselines in Mireshghallah et al. (2022).

### 4.2.1 EXPERIMENT SETUP

**Target and Source Data** Our target tasks in this section are MNLI and SST-2, which have respectively the largest and smallest number of examples among the four tasks studied in previous work (Yu et al., 2021; Li et al., 2022c; Bu et al., 2022b; Mireshghallah et al., 2022). The numbers of training examples

($N$) in MNLI and SST-2 are 393K and 67K. The source data for GLUE tasks is the pre-training corpus of BERT (Devlin et al., 2019); It consists of a subset of Wikipedia and the entire Bookcorpus. The source dataset has approximately 3.5 billion tokens.

**Model Sizes** We use models from the BERT family (Devlin et al., 2019). We consider four different model sizes (5M, 10M, 25M, and 44M). Details of the models are in Appendix C. Following previous work (Xia et al., 2022), we do not include embedding matrices when computing the number of parameters of BERT models. For text classification tasks, the BERT embedding layer during inference is simply a lookup table.

**Data Selection** For MNLI and SST-2, we experiment with selecting varying numbers of tokens from the source data. The target numbers of pre-training tokens are 20M, 40M, 200M, 400M, 800M, 1200M, and 2000M. More complete implementation details on data selection are in Appendix C.

**Baselines** The baselines include pre-training on randomly selected source data and pre-training on all source data. There are two additional baselines for the 44M model. The first is directly fine-tuning DistillBERT (Sanh et al., 2019) with differential privacy. DistillBERT is distilled from BERT-base on the source data. The second is the best result in Mireshghallah et al. (2022). Mireshghallah et al. (2022) compress a DP fine-tuned BERT-base model using differentially private distillation or pruning. The architecture of the compressed models in Mireshghallah et al. (2022) and Sanh et al. (2019) are of the same architecture as the 44M model. Although our framework is compatible with the techniques in Mireshghallah et al. (2022) and Sanh et al. (2019), we include the two additional baselines to demonstrate that the proposed framework alone is a competitive approach for model compression in private learning.

**Private Learning** We adopt the setup in Mireshghallah et al. (2022). The privacy budget is $(4, 1/10N)$-DP. The hyperparameters for private learning are also documented in Appendix C.

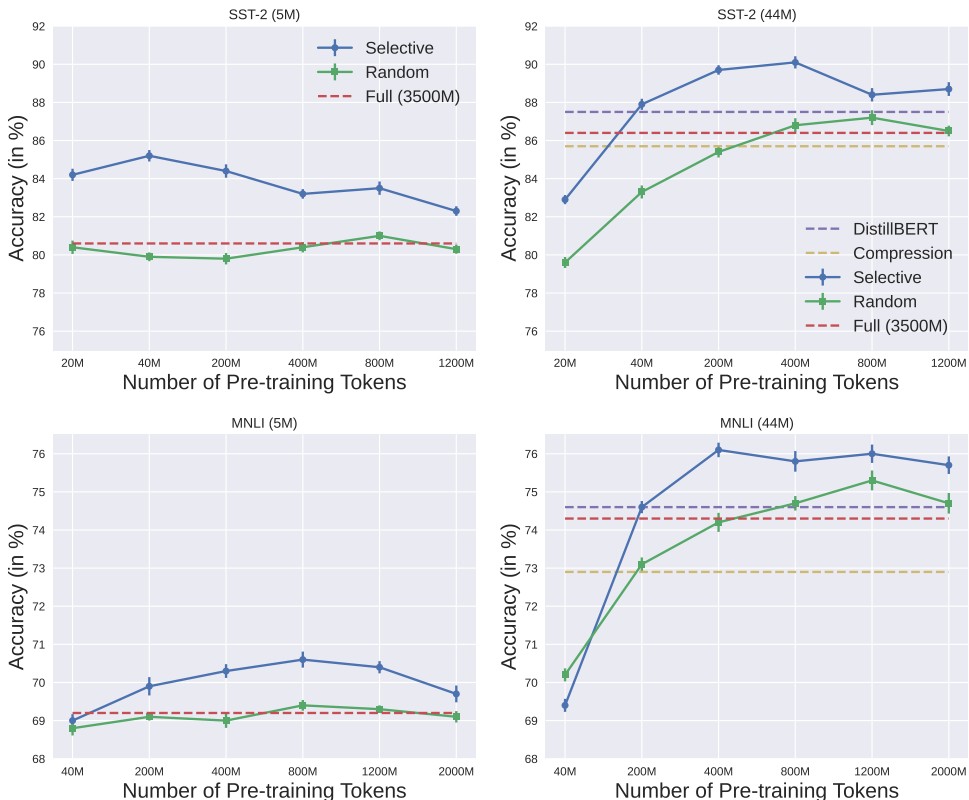

Figure 5: Results of pre-training with various numbers of tokens. The first column shows the results of 5M models and the second column shows the results of 44M models. Selective pre-training outperforms baseline algorithms in most of the cases.

### 4.2.2 SELECTIVE PRE-TRAINING OUTPERFORMS BASELINES, IMPROVES MODEL EFFICIENCY

Figure 5 shows the test accuracy on MNLI and SST-2 after privately fine-tuning models pre-trained with varying numbers of tokens. Our first finding is that, for most of the settings, selective pre-training outperforms all the algorithms examined. On SST-2, selective pre-training achieves accuracy that is 4.6% and 3.7% higher than the accuracy of full pre-training for the 5M and 44M models, respectively. On MNLI, the accuracy improvements are 1.4% and 1.8%, respectively. Considering the simplicity of these tasks, these improvements are non-trivial. Our second finding is that, for a model of fixed size, *increasing the number of pre-training tokens does not necessarily lead to better downstream accuracy.* This suggests that there may be an optimal number of pre-training tokens for a given model size (Sorscher et al., 2022; Hoffmann et al., 2022), further emphasizing the need to choose a task-specific subset from a large source data.

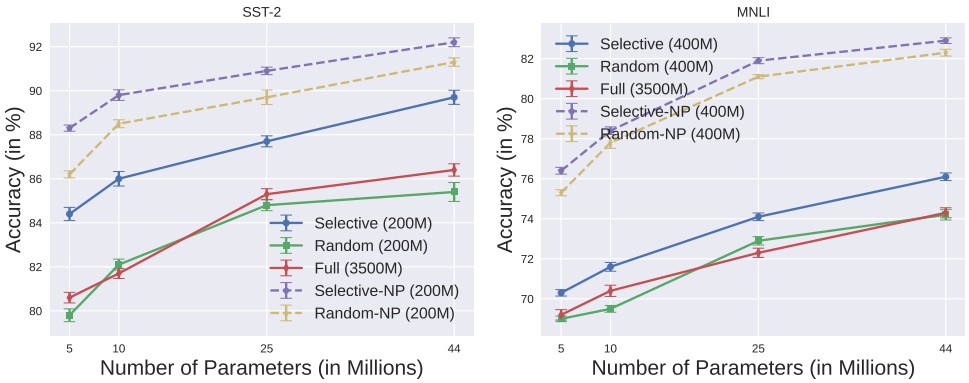

Figure 6: Results of pre-training with different model sizes. The numbers in the brackets are the numbers of tokens used for pre-training. 'NP' denotes that the models are fine-tuned without DP. Selective pre-training consistently improves performance across all settings. The improvements for models trained with DP are larger.

Figure 6 shows the test accuracy of models of different sizes. When trained with differential privacy, the 25M model with selective pre-training achieves comparable or better performance than the 44M baseline models, aligning with our observations on the Enron email dataset. The accuracy gains on SST-2 are greater than those achieved on MNLI, likely because MNLI data distribution is relatively closer to Wikipedia corpus; see Appendix B.1 for the word clouds comparison.

## 5 RELATED WORK

Our work is related to model compression, data-selection for pre-training (Xie et al., 2023; Gunasekar et al., 2023), and domain adaptation (Wang et al., 2020; Zhang and Gao, 2022; Bassily et al., 2023) literature, and we give a comprehensive summary and comparison with other works in Appendix A.

Closer to our work are the concurrent works of Hou et al. (2023) and Gu et al. (2023) that study how to privately select an optimal public dataset from an explicitly given list of public datasets. For instance, suppose the private dataset is CIFAR-10, and available public datasets are MNIST, CIFAR100, and ImageNet. The goal is to design a private algorithm to find which of the three public datasets is better suited for private learning on CIFAR-10. In this paper, we explore how to select a subset of a single public dataset on a sample-by-sample basis. Our algorithm does not require any explicit division of public data and runs efficiently on billions of tokens, making it well-suited for finding the right pre-training data for language models. More importantly, our emphasis is not just on model accuracy, but on how pre-training impacts accuracy-vs-model size trade-offs.

## 6 CONCLUSIONS, OPEN QUESTIONS, AND LIMITATIONS

A limitation of our work is that we provide DP guarantees only with respect to private datasets and not public datasets. In building applications, it is important to take into account the privacy risks of public data Tramèr et al. (2022). This work initiated the study of pre-training strategies that are friendly for

model compression. Immediate follow-up questions are: are there better algorithms for data selection? How should one select the pre-training data for general purpose models? Our work also touches upon scaling laws (Sorscher et al., 2022; Hoffmann et al., 2022), and understanding the scaling behaviour in private deep learning is a fascinating research direction (Sander et al., 2022). Finally, private learning as a tool for model efficiency can have a huge impact on real-world applications (MosiacML; ScaleAI), and more research in this space can lead to a better understanding of its power.

## REPRODUCIBILITY STATEMENT

We have submitted our source code through OpenReview and will make it public after the review process. Details regarding our hyperparameter choices can be found in Appendix C. All the datasets used in our experiments are publicly available.

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

## A   RELATED WORK: PUTTING OUR FRAMEWORK IN CONTEXT

For general literature on private deep learning and fine-tuning we refer the readers to (Abadi et al., 2016; He et al., 2023; Kerrigan et al., 2020; Li et al., 2022c; Bu et al., 2021; Lee and Kifer, 2021; Subramani et al., 2021; Anil et al., 2022; Yu et al., 2022; De et al., 2022; Mehta et al., 2022; Yu et al., 2021; Zhu et al., 2020; Sander et al., 2022; Bu et al., 2022a; Panda et al., 2022), and references there in. To the best of our knowledge, no prior work in DP literature has studied selective pre-training from scratch and its impact on the transfer learning abilities of a model. Our work is at the intersection of several related topics, and we give a brief overview of how our work fits into the broader literature on the topic.

**Domain Adaptation and Reducing Distribution Disparity Between Public and Private Data**
Public data has been widely used to improve private data analysis (Papernot et al., 2017b; Alon et al., 2019; Bassily et al., 2020b;a; Kairouz et al., 2021; Liu et al., 2021a; Zhou et al., 2021; Liu et al., 2021b; Amid et al., 2022; Yang and Cheng, 2022; Li et al., 2022a; Bie et al., 2022). To address the distribution shift between private and public data, a recent line of research explores domain adaption (Wang et al., 2021; Zhang and Gao, 2022) in the context of private learning (Wang et al., 2020; Zheng et al., 2023; Bassily et al., 2023). However, these works are not applicable to our setting due to many reasons, but in particular that we are interested in how pre-training dataset affects the model size. Much of the above literature considers simply the performance of the final model. In the absence of the model size restrictions, for NLP applications, it is well established He et al. (2023) that pre-training on large corpus of text using a large model offers better utility-vs-privacy trade offs.

**Non-Private Data Selection** Automatic data selection and cleaning, along with how the pre-training data impacts the downstream task performance are important problems in deep learning. See Xie et al. (2023); Gururangan et al. (2020); Brown et al. (2020); Chowdhery et al. (2022); Jain et al. (2023); Hernandez et al. (2022); Mindermann et al. (2022); Lee et al. (2022); Coleman et al. (2020) and references there in. Yet, the literature is scarce on the impact of selective pre-training on the model, except the recent concurrent work of Xie et al. (2023). Our work explores these questions in the context of private learning, with an emphasis on how the quality of data affects performance and model size. As a pilot study on designing privacy-preserving data selection algorithms, we use simple classification-based approaches that are easy to privatize and provide a clear illustration of the main messages of the paper. Exploring more sophisticated approaches Xie et al. (2023) for private data selection is an interesting future direction.

**DP Model Compression** The setting considered in this paper, where we want to train a model of a certain size, is related to the model compression literature, which was recently studied by Mireshghallah et al. (2022) in the private learning context. See references there-in for more DP compression work that is not directly related to our setting such as Papernot et al. (2017a; 2018); Lyu and Chen (2020); Tian et al. (2021). They use black-box compression techniques, such as distillation (Hinton et al., 2015) or pruning (Han et al., 2015), at the *fine-tuning* stage. The experiments in Section 4.2 show that using our framework alone can improve upon their results. Moreover, our framework is compatible with existing model compression techniques (Hinton et al., 2015; Han et al., 2015). How those techniques can be combined with selective pre-training to further improve the model performance and/or improve the compression ratio is an interesting research direction.

## B   MORE EXPERIMENTS

### B.1   RESULTS OF DATA SELECTION FOR GLUE TASKS

We plot the word clouds of SST-2/MNLI and (selected) source data to further demonstrate that the distribution of selected data is closer to the distribution of target data. The source data for SST-2 and MNLI is a subset of Wikipedia and the entire Bookcorpus.

The domain classifiers of SST-2 and MNLI are trained the same way as illustrated in Section 3. We select 400M tokens for SST-2 and MNLI, separately. The word clouds of the most frequent 100 nouns are in Figure 7 and 8. We exclude common prepositions and verbs in the word clouds. On SST-2, our selection algorithm improves the number of overlaps between the source data and the target data from 25 to 40. On MNLI, our algorithm improves the number of overlaps from 44 to 51.

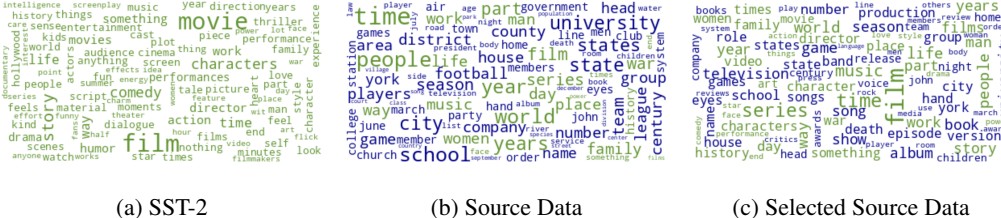

|     (a) SST-2     |     (b) Source Data     |     (c) Selected Source Data     |

Figure 7: The 100 most frequent nouns in SST-2, the source data, and a selected subset of source data. The source data is Wikipedia and Bookcorpus. Green words are the 100 most frequent nouns in SST-2. The source data and the selected subset have 25 and 40 words, respectively, that are among the 100 most frequent nouns in SST-2.

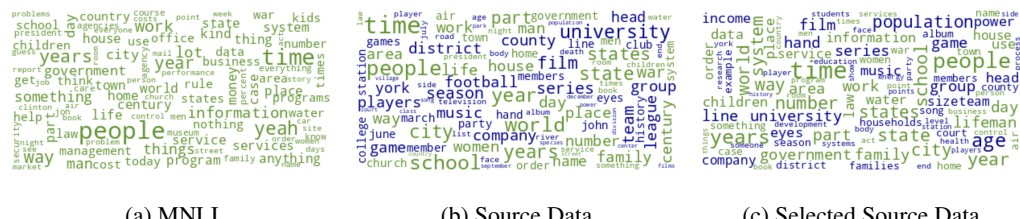

|     (a) MNLI     |     (b) Source Data     |     (c) Selected Source Data     |

Figure 8: The 100 most frequent nouns in MNLI, the source data, and a selected subset of source data. The source data is Wikipedia and Bookcorpus. Green words are the 100 most frequent nouns in MNLI. The source data and the selected subset have 44 and 51 words, respectively, that are among the 100 most frequent nouns in MNLI.

The results explain our findings in Section 4.2 that selective pre-training yields larger performance improvements on SST-2 than on MNLI.

Table 1: Next word prediction accuracy (in %) of GPT models on the Enron email dataset. The overall privacy budget is $(7.3, 1 \times 10^{-7})$.

| Parameters | 21M | 45M | 82M | 124M | 355M |
|---|---|---|---|---|---|
| Random | $32.8_{\pm 0.02}$ | $35.0_{\pm 0.01}$ | $36.5_{\pm 0.01}$ | $37.2_{\pm 0.02}$ | $38.4_{\pm 0.02}$ |
| Top | $33.8_{\pm 0.03}$ | $36.1_{\pm 0.02}$ | $37.5_{\pm 0.04}$ | $38.4_{\pm 0.02}$ | $39.4_{\pm 0.01}$ |

### B.2 MORE EXPERIMENTS ON THE ENRON EMAIL DATASET

Table 1 shows the top-1 next word prediction accuracy on the test split of the Enron email dataset as well as the standard deviation over five random seeds. With selective pre-training, a 82M model achieves an accuracy of 37.5% which is 0.3% higher than the accuracy of a 124M model that is not carefully pre-trained.

We also test selective pre-training under different privacy budgets. Figure 9 presents perplexity and next-word prediction accuracy of 21M and 355M GPT models under a wide range of $\varepsilon$ (ranging from 2.3 to 10.9). We fix the privacy parameter $\delta$ as $1 \times 10^{-7} < 1/10N$. We found that selective pre-training leads to similar improvements across all the choices of $\varepsilon$.

## C   IMPLEMENTATION DETAILS

This section expands on the implementation details that are omitted from the main text due to space constraints.

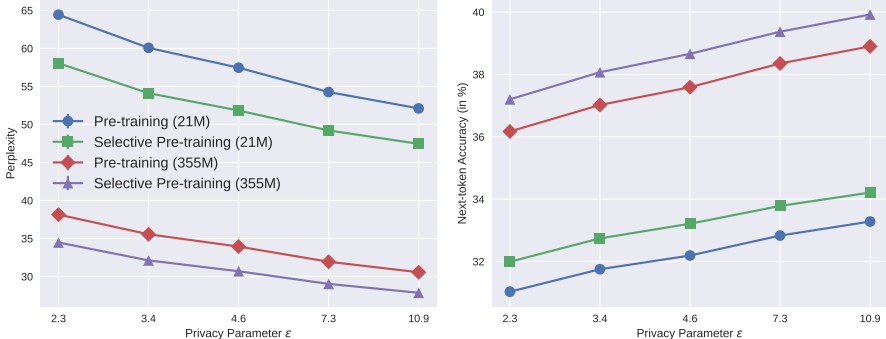

Figure 9: Perplexity and top-1 next-word accuracy on the Enron email dataset. We consider a wide range of $\varepsilon$ (ranging from 2.3 to 10.9). The numbers in brackets are the number of model parameters. The privacy parameter $\delta$ is $1 \times 10^{-7}$. Selective pre-training yields consistent gains across all $\varepsilon$ evaluated.

| Param. | 21M | 45M | 82M | 124M | 355M |
|---|---|---|---|---|---|
| $L$ | 4 | 4 | 6 | 12 | 24 |
| $d$ | 312 | 576 | 768 | 768 | 1024 |
| $d_{FFN}$ | 1248 | 2304 | 3072 | 3072 | 4096 |

| Param. | 5M | 10M | 25M | 44M |
|---|---|---|---|---|
| $L$ | 4 | 6 | 6 | 6 |
| $d$ | 312 | 384 | 576 | 768 |
| $d_{FFN}$ | 1200 | 1200 | 2304 | 3072 |

Table 2: Architecture hyperparameters of the models for the Enron email dataset.

Table 3: Architecture hyperparameters of the models for GLUE tasks.

**Details of the Models** Let $L$, $d$, and $d_{FFN}$ be the number of layers, hidden size, and intermediate size of the fully connected block, respectively. We change $L$, $d$, $d_{FFN}$ to get different model sizes. Other architecture hyperparameters are the same as those in Devlin et al. (2019) and Radford et al. (2019). Table 2 and 3 show the model details for the Enron email dataset and GLUE tasks, respectively.

**Data Selection for Enron Email** The text in OpenWebText is also divided into sequences of length 256. To construct the training set of the domain classifier, we randomly sample $5N$ sequences from the source data as negative samples, and use all $N$ sequences in the target dataset as positive samples. As a result, the training set of the domain classifier is 6 times larger than the target data. This significantly reduces the privacy cost of training the domain classifier because the probability of a target example being sampled becomes 6 times smaller. We initialize the domain classifier with an 82M GPT model pre-trained on OpenWebText and fine-tune it with DP-Adam on the constructed training set.

**Data Selection for GLUE Tasks** Because the positive examples in SST-2 and MNLI are natural sentences instead of sequences of fixed length, we sample natural sentences in the source data as negative examples for training the domain classifier. The domain classifier is initialized with BERT-base. In MNLI, a single example contains two natural sentences, i.e., a premise and a hypothesis. In this case, only one of the two sentences is chosen randomly as a positive example. The number of negative examples is also $5N$.

The pre-training sequences in Devlin et al. (2019) are of a fixed length. Each sequence may consist of several natural sentences. To get the ranking score of a sequence, we first break a fixed-length sequence into natural sentences and use the domain classifier to predict those sentences. The maximum confidence of the sentences is used as the ranking score for the sequence.

**Hyperparameters For Pre-training** The pre-training process uses common hyperparameters in the literature. For pre-training models from the BERT family, we follow the hyperparameters in Devlin et al. (2019). The hyperparameters for pre-training models from the GPT family are as follows. We use a dropout probability of 0.1 and a weight decay of 0.01. The $\beta_1$ and $\beta_2$ of Adam are 0.9 and

Table 4: Hyperparameters for private fine-tuning. We use $N$ to denote the size of the target dataset.

| Pre-training Method | Standard | Selective |
|---|---|---|
| Noise multiplier (Enron) | 1.00 | 1.03 |
| Noise multiplier (SST-2) | 1.36 | 1.38 |
| Noise multiplier (MNLI) | 1.44 | 1.46 |
| Train steps (domain classifier) | N/A | 100 |
| Train steps (target task) | [150, 500, 1000] | |
| Clipping norm | 1 | |
| Learning rate | [1e-4, 5e-4, 1e-3, 3e-3] | |
| Weight decay | 0 | |
| Batchsize | $\lfloor 0.03N \rfloor$ | |
| Privacy budget | $(7.3, 1 \times 10^{-7})$ for Enron; $(4, 1/10N)$ for GLUE | |

0.999, respectively. All models are pre-trained from scratch for 100K iterations with a batch size of 128. The initial learning rate is $5 \times 10^{-4}$ and follows a linear decay schedule.

**Hyperparameters For Private Fine-tuning**   We follow the findings in previous work to set most of the hyperparameters (Li et al., 2022c; Mireshghallah et al., 2022). We additionally tune the learning rate to adapt to the various model sizes we studied. Table 4 summarizes the hyperparameters for private learning. We use the parameter-efficient fine-tuning algorithm LoRA (Hu et al., 2022) to improve the efficiency of the DP fine-tuning of GPT models Yu et al. (2022). We do not use LoRA for the DP fine-tuning of BERT models to get a fair comparison to Mireshghallah et al. (2022). For a given set of hyperparameters, we use the PRV accountant to get the noise multiplier of DP-Adam. If we use selective pre-training, then the noise multiplier is slightly larger because we need to account for the privacy cost of training a domain classifier. We repeat each private fine-tuning experiment 5 and 3 times with different random seeds for the Enron email dataset and GLUE, respectively.

