# OpenReview forum: "Differentially Private Model Compression via Selective Pretraining"
_ICLR.cc/2024/Conference — Submitted to ICLR 2024_

### Official Review · Reviewer_mFxj · 2023-10-26

**Soundness:** 4 excellent
**Presentation:** 4 excellent
**Contribution:** 4 excellent
**Rating:** 8
**Confidence:** 4

**Summary:**

The rise of expansive, publicly accessible language models has transformed the AI landscape. These models can perform impressively, but their deployment in privacy-sensitive contexts raises concerns due to data sensitivities. As a remedy, there's growing interest in training these models using differential privacy (DP). However, the trade-off is stark: as data volume increases, ensuring privacy becomes costlier due to the need to introduce more noise. An emerging approach to this challenge is DP training of large models by fine-tuning them on limited private data—a method that's proving to be highly effective. But a significant obstacle remains: the resulting models, while powerful, are often too large and inefficient for real-world deployment, especially in systems designed to serve vast user bases. This limitation has spurred research into model compression, which seeks to encapsulate the capabilities of these models without compromising utility. This study focuses on fine-tuning language model, initially pre-trained on a selectively chosen subset of publicly available data, on private data using differential privacy methods. This paper leverages insights from private data to judiciously curate a subset of public data for pre-training purposes. The proposed framework unfolds in three phases: firstly, a privacy-preserving algorithm selects a subset of public data for pre-training (referred to as selective pre-training). This is followed by non-private pre-training on this chosen public data. The final phase involves private fine-tuning using the standard DP-SGD methodology.

**Strengths:**

- **Innovative Data Selection**: The paper introduces a novel approach of leveraging private domain knowledge to judiciously select subsets of public data for pre-training. This methodology ensures relevant data selection, enhancing the model's effectiveness in domain-specific applications.

- **Enhanced Performance**: The strategies employed lead to notable performance gains, demonstrating the efficacy of the proposed methods in comparison to conventional techniques.

- **Optimized Model Size**: One of the standout achievements of this work is the successful reduction in model size. This not only makes the model more resource-efficient but also more feasible for deployment in real-world scenarios with resource constraints.

- **Empirically Supported Findings**: The experiments presented in this work are thorough, and claims made by the authors are substantiated with appropriate evidence.

**Weaknesses:**

Notable points that can enhance the paper and improve readers comprehensibility:

- In the paper, PRV accountant was chosen for the DP analysis. Could the authors elaborate on the rationale behind this selection, especially considering the moment accountant is often a more prevalent choice in similar works? Understanding the specific advantages or reasons for this choice would offer greater insight into the methodology.

- In Section 4.1.1, the authors provided details about the size of the token but the size of the sequence for OpenWebText isn't specified. For clarity and consistency, it would be beneficial to include the sequence size for OpenWebText, especially since the sequence size for the target dataset is mentioned. This addition will offer a more direct comparison and help readers understand the relationship between the two datasets more effectively.

- I observed that Figure 1 and Figure 4 have variations in the y-axis and gridline spacing. To enhance the clarity and ease of comparison between the two figures, it might be beneficial to standardize the grid spacing across both. This consistency would aid readers in drawing more direct comparisons between the charts."

- In Section 1.1(Differential Privacy as a Tool For Model Compression), the authors mention, "Observe that the performance of a tiny model with 21 million parameters can match the zero-shot performance of GPT2-XL public model with 1.5 billion parameters." Upon reviewing the referenced Figure, it appears that the tiny model's performance is slightly lower in terms of accuracy (approximately 33.9% for Sel. PT+DP as opposed to 35.1% for GPT-XL) and exhibits a higher perplexity (around 49 for Sel. PT+DP compared to 44 for GPT-XL). Could the authors confirm if my interpretation aligns with your findings? While the tiny model doesn't precisely "match" (i.e 33.9% != 35.1%) the GPT-XL's performance, it still delivers a commendable performance considering the vast difference in parameter size (21 Million versus 1.5 Billion). It might be beneficial for clarity if the authors could consider revising this section to reflect these nuances.

**Questions:**

- In Section 1.2, could you specify the particular English corpus used to train the EnglishGPT-124M model? Additionally, for the EnglishGPT-82M's training data, is it a subset of the aforementioned corpus, or is it an entirely distinct dataset that bears distributional similarities to the Enron dataset?


- Could the authors expatiate on how they have defined the unit of privacy protection in this work? Specifically, is each sequence treated as a single datapoint such that its addition or removal wouldn't impact the algorithm's output? If so, does the sequence length have an influence on the privacy budget? In other words, would a longer sequence length entail a higher privacy cost?

- "In Appendix C, when discussing the Data Selection for the Enron Email, there seems to be a mention of the data being "6 times larger" than the target data (likewise 6 times smaller). However, in Section 3.1, it's indicated that negative examples are "five times larger than the number of positive examples". Could there be a discrepancy between these two sections? It would be helpful for clarity if these numbers are consistent throughout the paper."

---

> ### Author Response · Authors · 2023-11-21
> **Rebuttal: Our answers**
>
> We thank the reviewer for the encouraging and constructive feedback. We have taken your suggestions to improve our submission. Please find our response below.
>
> Q: PRV accountant was chosen for the DP analysis. Could the authors elaborate on the rationale behind this selection.
>
> A: We use the PRV accountant because it provides tighter bounds on privacy parameters $\varepsilon$ and $\delta$ than moments accountant.  We have added an explanation regarding this choice in Section 2.
>
> Q: For clarity and consistency, it would be beneficial to include the sequence size for OpenWebText.
>
> A: The sequence size for OpenWebText is 512, which aligns with the sequence length mentioned in the original GPT paper [1]. This is shorter than the sequence length in the GPT-2 paper [2] (1024). We set the sequence size to be 512 to reduce the computational expenses. As per your suggestions, we have added these details in the revision.
>
> Q: Figure 1 and Figure 4 have variations in the y-axis and gridline spacing.
>
> A: Thanks for the suggestion. We will make the y-axis spacing in Figure 1 and Figure 4 to be the same. For displaying the perplexity, both figures now use a spacing of 2. For displaying the accuracy, both figures now use a spacing of 1.
>
> Q: It appears that the tiny model's performance is slightly lower in terms of accuracy (approximately 33.9% for Sel. PT+DP as opposed to 35.1% for GPT-XL) and exhibits a higher perplexity (around 49 for Sel. PT+DP compared to 44 for GPT-XL).
>
> A: We thank the reviewer for the careful reading. Yes, your interpretation aligns with our findings, and it was a typo. We have revised this paragraph to the following. “Observe that the
> performance of **a small model with only 45 million parameters is comparable to** the zero-shot performance of GPT2-XL public model with 1.5 billion parameters.”
>
>
> Q: In Section 1.2, could you specify the particular English corpus used to train the EnglishGPT-124M model? Additionally, for the EnglishGPT-82M's training data, is it a subset of the aforementioned corpus, or is it an entirely distinct dataset that bears distributional similarities to the Enron dataset?
>
> A: The pre-training corpus for the EnglishGPT-124M model is the full OpenWebText. The EnglishGPT-82M’s pre-training data is a subset of OpenWebText,  which is selected using the algorithm in Section 3.1.
>
>
> Q: Specifically, is each sequence treated as a single datapoint such that its addition or removal wouldn't impact the algorithm's output? If so, does the sequence length have an influence on the privacy budget?
>
> A: For the GLUE tasks, the unit of privacy protection is a single training sample in the original datasets. For the Enron email datasets, we break the dataset into sequences of length 256 and the unit of privacy protection is a single sequence (we discussed this design choice in Section 4.1.1).
>
> This design choice is based on the observation that most of the emails in the Enron email dataset are shorter than 256. For long emails, it is possible that it may be splitted into multiple training sequences, and hence the privacy budget for that email would increase. Another option is simply to use a random subset of 256 contiguous tokens. However, we agree that in real world applications, it is important to carefully bound the maximum contribution of a single email/user.
>
> Thanks for your suggestion, and we will add this discussion to our revision.
>
>
> Q: In Appendix C, when discussing the Data Selection for the Enron Email, there seems to be a mention of the data being "6 times larger" than the target data (likewise 6 times smaller). However, in Section 3.1, it's indicated that negative examples are "five times larger than the number of positive examples".
>
> A: Thanks for the question. Our description in Appendix C is consistent with our description in Section 3.1. The training set of the domain classifier contains 6N samples, where N is the size of the target data. The training set contains 5N negative samples and N positive samples, which is 6 times larger than the size of the target data. We will revise the corresponding descriptions to clarify this.
>
>
> [1]: [Improving Language Understanding by Generative Pre-Training](https://cdn.openai.com/research-covers/language-unsupervised/language_understanding_paper.pdf)
>
> [2]: [Language Models are Unsupervised Multitask Learners](https://d4mucfpksywv.cloudfront.net/better-language-models/language_models_are_unsupervised_multitask_learners.pdf)

---

> > ### Comment · Reviewer_mFxj · 2023-11-22
> > **Response**
> >
> > I thank the authors for their response. I have no additional comments at this time. My assessment score remains unchanged.

---

### Official Review · Reviewer_CqKu · 2023-10-29

**Soundness:** 3 good
**Presentation:** 4 excellent
**Contribution:** 3 good
**Rating:** 6
**Confidence:** 4

**Summary:**

This paper presents a novel framework for training differentially private language models with a focus on model compression and efficiency. The authors propose a selective pre-training approach, which involves using a privacy-preserving algorithm to select a subset of public data for pre-training. This is followed by private fine-tuning via differentially private stochastic gradient descent (DP-SGD). The main contributions of the paper are:

1. A new framework for training domain-specific language models with differential privacy, which includes selective pre-training and private fine-tuning.
2. Demonstration that selective pre-training is crucial for smaller models to achieve better performance when privately fine-tuned with DP-SGD.
3. State-of-the-art results on standard NLP benchmarks, outperforming previous methods in the literature.
4. An empirical evaluation showing that smaller models trained with the proposed framework can match the performance of much larger models without access to private data, highlighting the promise of private learning as a tool for model compression and efficiency.

The paper also discusses the real-world impact of the proposed framework, as it has been used to train an industry-grade differentially private text prediction language model that serves many NLP applications in a large AI company.

**Strengths:**

The authors have made several concrete contributions in their paper, which can be assessed across the following dimensions:

1. Originality: The authors introduce a novel framework for training differentially private language models, focusing on model compression and efficiency. They propose a selective pre-training approach, which involves using a privacy-preserving algorithm to select a subset of public data for pre-training. This approach has not been explored in the context of differentially private learning before, and it represents a creative combination of existing ideas and techniques.

2. Quality: The paper is well-written and presents a clear and coherent argument. The experiments are well-designed and rigorously conducted, with appropriate baselines and a thorough analysis of the results. The authors demonstrate that their framework achieves state-of-the-art performance on standard NLP benchmarks, outperforming previous methods in the literature.

3. Clarity: The paper is well-structured and easy to follow, with clear explanations of the proposed framework, the experimental setup, and the results. The authors use appropriate figures and tables to illustrate their findings, making it easy for readers to understand the key points. The paper also provides a detailed description of the implementation details, which is helpful for reproducing the experiments.

4. Significance: The proposed framework has the potential to significantly advance the field of differentially private learning, particularly in the context of language models. By demonstrating that selective pre-training can lead to smaller models that match or surpass the performance of larger models without access to private data, the authors highlight the promise of private learning as a tool for model compression and efficiency. This has important implications for real-world applications, where model size and inference time are critical factors.

Overall, the paper is an original and significant contribution to the field of differentially private learning, with high quality and clarity. The proposed framework has the potential to advance the state-of-the-art in training efficient and high-performing private language models, which is a key challenge in the field.

**Weaknesses:**

While the paper presents a novel framework for training differentially private language models and achieves state-of-the-art results, there are a few areas where the work could be improved:
1. **Privacy guarantees:** The paper provides differential privacy guarantees only with respect to the private dataset and not the public dataset. It would be beneficial to extend the analysis to consider the privacy risks of the public data as well. This could involve incorporating privacy amplification techniques or exploring the composition of privacy guarantees for the entire framework.
2. **Analysis of scaling behavior:** The paper touches upon scaling laws in private deep learning, but a more in-depth analysis of the scaling behavior of the proposed framework would be beneficial. This could involve studying how the performance and efficiency trade-offs change with model size, data size, and privacy parameters.
3. **Generalization concerns**: As the theoretical analysis is lacking, it is reasonable to be concerned about the generalization ability of the proposed method. It would be beneficial if the authors could theoretically or empirically (i.e., with more evidences)  show its generalization ability.
By addressing these areas, the paper could provide a more comprehensive and robust framework for training differentially private language models, ultimately leading to more efficient and high-performing models.

**Questions:**

Overall, this method is interesting, simple, and effective (overwhelming SOTAs). I have the following suggestions:
1. **Veritable privacy protection**: It will be interesting to discuss with authors about *what is the real privacy of the private dataset*. In the proposed method, a selected dataset is set as a proxy of private dataset. Although the selection is protected via DP, the resulting proxy dataset is *similar* to the private dataset. In this case, is the DP sitll meaningful?
1. **Comparison with other model compression techniques**: The paper shows that the proposed framework can improve upon existing model compression techniques when used alone. However, it would be interesting to explore how the framework can be combined with other compression techniques, such as knowledge distillation or pruning, to enhance model performance and compression ratios further.
2. **Data selection algorithms**: Are there any plans to explore more sophisticated data selection algorithms, such as those based on importance resampling or data pruning? How do these alternative methods compare to the simple classification-based approach used in the paper in terms of model performance and efficiency?
3. **Real-world applications**: Can the authors provide more details on the real-world impact of the proposed framework, such as the specific NLP applications it has been used for and the performance improvements observed in these applications?
4. **Reproducibility:** Can the authors provide more details on the reproducibility of the experiments, such as the exact code and data used, as well as any potential pitfalls or challenges that other researchers may encounter when attempting to reproduce the results?

---

> ### Author Response · Authors · 2023-11-21
> **Rebuttal Part 1: Answers to specific questions**
>
> Thanks for the detailed and fair review. We appreciate your time and effort, and we will certainly incorporate some of your feedback in the final version of the paper. Below we address your specific questions.
>
> Q: Privacy guarantees of public data and generalization concerns.
>
> A: These are great questions. Consider our experiments on the Enron email dataset. To ensure that there is no data contamination in the selected pre-training data, we did a 10-gram deduplication between the Enrom emails and the full OpenWebText before the selection. The deduplication follows the algorithm described in Section 4 of the GPT-2 paper [1].
>
> Conceptually, the improvement of our algorithm could be interpreted as follows. Writing styles of users vary according to the context. For example, writing emails is different from writing academic papers which is different from language usage in text messages. While particular emails of users are private, the use of language in writing emails among the population has some common “stylistic similarities” that are preserved by data selection algorithms. Putting it another way, our hypothesis is that, while emails themselves are private, the style and language usage in writing emails is not.  The goal of selective pretraining is to first teach this style of writing emails to the model.
>
> Our industry deployments in multiple applications also gives confidence that our framework generalizes to more real-world settings, and this generalization is not due to examples similar to downstream datasets being there in public datasets.
>
> Q: More experiments on scaling behavior.
>
> A: We agree with you that more experiments are needed to understand the scaling behavior of the models. This is one of the main reasons we did not emphasize this aspect as much as we wish to. Performing experiments to establish scaling behavior (in a spirit similar to the Chinchilla paper) is extremely expensive, and we hope to do that in the future. Unfortunately, however, we may not be able to do that for this paper.  But we think that this is a great research direction and one of our fav open problems.
>
> Q: Comparing with other compression techniques.
>
>
> We agree that our framework can be combined with other compression techniques. However, as has been shown by Mireshghallah et al, a black box application of Knowledge Distillation algorithm in DP setting is challenging and can lead to huge performance loss. On the other hand, pruning techniques typically lead to unstructured sparsity, which does lead to better inference time. These methods were explored in Mireshghallah et al paper.
>
> Having said that, one possible approach to further compress our models is to take our selectively pretrained language model, do model compression using the non-private KD algorithm on the public data, and then followed by DP fine-tuning on the private data. It is reasonable to believe that this could further improve compression ratios.
>
> Goal of this paper was to initiate these conversations, and to show that private data can be used at all stages of the LLM training pipeline in novel ways. While performing all the experiments you suggest may not be possible within the scope of this paper, we hope that our work sets benchmarks that others can improve upon. Moreover, our work already gives enough evidence to the main technical question we set forth to answer: Can small DP models achieve good utility vs privacy tradeoffs as that of larger ones, which was an intriguing open problem in this space.
>
>
> Different Data Selection Policies:
>
> This is a great question.  We agree that more sophisticated data selection policies may lead to better results. However, we chose a simple classifier-based approach for two reasons: 1) Our approach needs no extra work to make data selection step DP, is simple and effective. 2) We wanted to show the power of the overall framework rather than the data selection algorithm itself. We hope that our work gives motivation to explore more sophisticated data selection algorithms that you mention, and study them in the context of DP.
>
> Reproducibility:
>
> Our source code is in the supplementary, which contains a detailed readme to reproduce the main findings in the submission. We will also make the source code available online as soon as the review process is over. We are happy to take any further suggestions regarding our code.
>
> [1]: [Language Models are Unsupervised Multitask Learners](https://d4mucfpksywv.cloudfront.net/better-language-models/language_models_are_unsupervised_multitask_learners.pdf)

---

> > ### Author Response · Authors · 2023-11-21
> > **Rebuttal Part 2: Details about the real world impact**
> >
> > Our framework was used to train a 30 million parameter transformer model to predict words/sentences in email and note taking applications. We do not want to give names of these applications to preserve anonymity of the review process. The model trained with our framework matched (slightly outperformed) a much larger public model with 120 million parameters.  Figure 1, which compares the perplexity and next word prediction accuracy of our models with the public model, is a fair approximation of the performance improvements.
> >
> > From a deployment standpoint, what stood out more than the absolute performance gains in terms of perplexity and prediction accuracy, is the decrease in inference cost and time. Our model decreased the inference cost by a factor of 47% ($1.9 to $1 Cost Per Million requests). Since our model serves 20 billion queries per day, this is savings in millions of dollars per year.
> >
> > Now, let us discuss how decrease in the inference time improves performance metrics such as number of accepted suggestions/characters saved (improvement of 17% in  A/B testing), which can be very different from improvements in perplexity/accuracy.  The text prediction engine in our email client and note taking app has the following architecture.  When a user is typing, the engine sends a query to the cloud to predict the next word/sentence. Simultaneously, it starts a timer. If the server does not respond back by that time, it uses a simple heuristic to predict the next word or offers no suggestion. If the server responds back before the timer expires, then it shows the predictions of the server.
> >
> > In such a framework, inference latency is as important as prediction accuracy. If the server does not respond back within the given time, prediction accuracy becomes irrelevant.
> > As our model is 1/4th size of the previous one, the inference time also improved significantly.
> >
> > Finally, this is the largest deployment of a DP transformer model based on our knowledge.
> >
> > These experiments also give us confidence that our framework can generalize to various other settings.
> >
> > Let us know if you need any further clarifications. If you are happy with our answers, please do consider raising the score. We are quite excited by this work, and we believe that some of the questions you asked us would be great future research directions. Hence, a venue like ICLR is befitting to disseminate these ideas to a broader DP audience.

---

> > > ### Comment · Reviewer_CqKu · 2023-11-22
> > > **Response to Authors' Rebuttal**
> > >
> > > Thanks to the authors for their careful response. Indeed, some of the suggestions are expensive to implement, while they are important to improve the overall quality of this research. I will keep my score.

---

### Official Review · Reviewer_kdmU · 2023-10-30

**Soundness:** 3 good
**Presentation:** 2 fair
**Contribution:** 2 fair
**Rating:** 5
**Confidence:** 4

**Summary:**

the paper discussed the impact of data selection in the pre-training step on the performance of relatively small language models, and proposed an approach that provides differential privacy guarantees to the finetuned small language models on downstream tasks. The proposed approach involves three steps:
1. privately select training data from a large corpus using a classifier optimized using DP-SGD.
2. non-privately pre-train a language model using the selected training data
3. privately finetune the model on the downstream task using DP-SGD.

( Also, thanks for correcting my misunderstandings, but now i have more practical concerns. )

**Strengths:**

1. the private classification model learns the distribution of the private data, which then selects similar texts from the public data for pre-training. It indeed improves the performance, and potentially reduces the latency during pre-training.

2. the empirical performance with a reasonably large eps value is decent.

**Weaknesses:**

1. the concept of pre-training is that the resulting model is generic enough that it is easy to finetune and adapt to a specific dataset, however, the proposed algorithm pre-trains a domain-specific model and then finetune it to a private dataset.

2. given that there are already many public pre-trained LLMs, wouldn't it make more sense to mix the selected public data and private data into a single finetuning step? and only do DP-SGD on the private portion of the data?

3. we have to keep in mind that eps means that the difference of the output distributions is bounded by exp(eps), and exp(10) is already a very large number, which practically doesn't provide much privacy protections. It would be better to see the performance or the impact when smaller eps values are used.

**Questions:**

n/a

---

> ### Author Response · Authors · 2023-11-21
> **Rebuttal about correctness, the proof, and clarifications**
>
> We thank the reviewer for careful reading of the paper and giving us valuable feedback. We appreciate your time and effort.
>
> It appears that there is a misunderstanding of our framework, which made you think that our DP guarantees are wrong. We can assure you that our framework is provably and rigorously (epsilon, delta)-DP with respect to the private data as defined in Section 2.
>
> Before we elaborate, let us first try to understand where the confusion is. From your comment, it appears that you expect DP guarantees to hold on the public data, a subset of which is selected for pretraining? If this is the case, then you are right that our framework does not provide any DP guarantees on the public data set. However, this is by our definition.
>
> Let us recall our problem setting as described in Section 2: Input to our problem is a private dataset D(priv) corresponding to a downstream task T, a model M of size p, privacy parameters epsilon > 0, delta > 0,  and a public dataset  D(pub). Our goal is to train  M on public and private datasets with the aim of maximizing the downstream performance on the task T. The entire process should be (epsilon, delta)-differentially private with respect to D(priv).
>
> Note that our problem definition does not require us to give privacy guarantees on the D(pub). Public dataset is assumed to be public, and this is the assumption in all the previous works in DP + LLM space, and nothing specific to this work. See for example: “Large language models can be strong differentially private learners ICLR 2022, “Exploring the limits of differentially private deep learning with group-wise clipping ICLR 2023”, and most of the works cited in our paper.
>
> Now if you are concerned that our framework is not DP with respect to D(priv),  and we are happy to provide a full proof in the final version of the paper. We omitted the proof and only gave a sketch in Section 2 for two reasons: a) The privacy analysis of DPSGD is somewhat standard and is well known in the community b) Due to page limit. To give you some more confidence, a  model trained using our framework is serving billions of queries per day in a top AI company, so we would not afford to make such a mistake. Having said that, we agree with you that providing a full proof helps to make our paper self contained. We will incorporate your feedback in the final version.
>
> Let us give a brief explanation of why our overall framework is differentially private with respect to D(priv).  First of all, finetuning stage of our framework uses DPSGD and thus it is (epsilon, delta)-DP with respect to D(priv).  Another step in our algorithm that uses private data is training the classifier. Our classifier is simply a 124-million parameter pretrained GPT2 model. We train it using DPSGD on the following dataset: 1) samples from D(priv) labeled as positive 2) some random subset of public data labeled as negative. The classifier is trained to predict positive labels. We train the model using DPSGD. This is explained in Section 3.1
>
> The proof of DPSGD (see for example Abadi et al 2016, Gopt et al 2021 ) says that *weights* of the classifier satisfies DP guarantees. This further implies that we can use this model to do data selection without any additional privacy loss due to the post processing theorem.
>
> We split the privacy budget to account for both these steps. We allocate a small fraction of the privacy budget to train the classifier and the rest of the privacy budget is allocated to the fine-tuning stage. Finally, we use advanced composition to calculate the final privacy loss. This is also explained in Section 3.1 and in the experiments section.
>
> We hope this alleviates your concern. We are happy to clarify if you have further questions.
>
> Multiple reviewers (cqKu,mFxj) found our work novel, important, and correct. We are quite excited about this work as 1) it shows that private data can play a role in all the stages of the training pipeline, and opens up exciting new research directions 2) one can train small language models of size 40 million parameters that are as good significantly larger public models, which was an important open problem in the literature. Further, since our model is now being deployed in a large AI company and serves billions queries per day, it gives further evidence that our framework should generalize to real life work loads.
>
> If you feel more confident about correctness now, please consider revising the score.

---

### Official Review · Reviewer_3s2h · 2023-10-31

**Soundness:** 2 fair
**Presentation:** 2 fair
**Contribution:** 1 poor
**Rating:** 3
**Confidence:** 4

**Summary:**

This paper introduces an approach to compress the size of LLMs on private data. The idea is to start a pretrain a small model from scratch and select the pretraining data smartly, i.e. close to the distribution of private data. To achieve this, the authors propose a method for selecting private data by training a binary domain classifier privately. This classifier distinguishes whether a given data point belongs to the target data distribution or not, allowing the selection of data points with higher confidence from the target domain. After the model is pretrained on the selected dataset, then it is fine-tuned on the private dataset. The efficacy of the proposed approach is demonstrated through experiments conducted on various text datasets, encompassing diverse data and model sizes.

**Strengths:**

- The topic is well-motivated, as it is important to study how to compress the LLMs for sensitive usecases where private learning is a must.
- The method presented is straightforward and easily comprehensible. The authors have provided empirical evidence demonstrating the superior performance of their approach compared to the baseline through an extensive and large-scale experimental setup.

**Weaknesses:**

- The title is a bit misleading as compression typically means compressing a larger model into a smaller one while this work focuses on how to better pretrain a smaller model from scratch.
- The technical innovation in this study is relatively constrained. The idea of training a classifier for selecting better data has been adopted in GPT3. Previous works, such as [1], have already introduced domain-specific pretraining, and techniques for data selection in training language models have been introduced in [2, 3]. The innovation of this work is the private learning of the domain classifier, which is limited.
- Some other baseline are missing: 1) if one knows the domain of the private task, then one could select pretraining data in that domain without having to perform the DP data selection. 2) zero-shot learning of compressed or quantized larger models.

Reference

[1] Gururangan, Suchin, et al. "Don’t Stop Pretraining: Adapt Language Models to Domains and Tasks." Proceedings of the 58th Annual Meeting of the Association for Computational Linguistics. 2020.

[2] Moore, Robert C., and William Lewis. "Intelligent selection of language model training data." Proceedings of the ACL 2010 conference short papers. 2010.

[3] Ruder, Sebastian, and Barbara Plank. "Learning to select data for transfer learning with Bayesian Optimization." Proceedings of the 2017 Conference on Empirical Methods in Natural Language Processing. 2017.

**Questions:**

- DP training is known for introducing worse calibration to the classifier. How might that impact the data selection process?
- One can also select the public dataset with DP to pretrain a large model, and then compress and finetune it on private data. How does this compare with pretraining a smaller model?

---

> ### Author Response · Authors · 2023-11-21
> **Rebuttal Part 1: Technical innovation**
>
> We thank the reviewer for careful reading of the paper and giving us valuable feedback. We appreciate your time and effort.  Before we address your specific questions, we want to offer a different view of our results and seek your feedback.  The review seems to evaluate the paper by directly comparing it to non-DP deep learning and concludes that technical novelty is limited. However, our main focus is on private language models and the role of private data. Even if we accept that all the ideas have been tried in non-private literature, which we do not completely agree with, it is not at all clear to us why these things should come together in the private world and lead to good DP models. To give an example, knowledge distillation works great in the non-private world, but we know it does not work in the DP world; see Mireshghallah et al. paper.
> Further, there is no evidence in the DP literature prior to our work to suggest that a 30 million parameter model can beat the next word prediction accuracy of a significantly larger public model and satisfy DP guarantees at the same time.  Moreover, we believe that our work introduces several novel ideas to the DP LLM community, which other reviewers (mFxj,cqKu) seem to agree with and find it novel.
>
> We list some novelty of our work with respect to DP Literature:
>
> Almost all the recent results in DP literature suggest that one needs large pretrained models to achieve good performance in the private world. See “Exploring the limits of DP DL with group wise clipping” for example. One of the main open problems in differentially private deep learning is to bridge the utility-vs-privacy gap between large language models compared to small language models. As you can verify, other reviewers agree with this and find our solution novel. This is the first paper to show that small models can achieve comparable utility-vs-privacy tradeoffs, and we achieve SOTA of the results for DP performance for small models improving upon Mireshghallah et al. [58].
>
> No prior work in DP literature has studied how one must do pretraining to enhance the transfer learning abilities of a model and its impact on the model size. Here we also show that better pretraining is more important for DP deep learning compared to non-private deep learning.
>
> Our work shows how the quality of tokens can change pretraining scaling behavior.
> If one is not careful in pretraining data quality, then we need a larger model to achieve the same performance. On the other hand, restricting pretraining dataset to smaller but better tokens, one can train a smaller model. We are not aware of such a study in DP literature before.
>
> We provide a general framework on how a small amount of private data can be used to improve pretraining, fine-tuning, and data selection. No prior work has explored utilizing private data at all stages of the training process.
>
> In many real-world applications, the public models such as GPT or LLaMA cannot be trained (or fine-tuned) on private datasets due to privacy concerns. Hence, both improvements in performance and model efficiency are possible if one had access to the private data. Privacy enhancing technologies such as DP provide the infrastructure to safely unlock the high-quality data, thus simultaneously improving both performance and model efficiency.
>
> DP as a tool for model compression and using private learning to improve efficiency vs accuracy tradeoff has never appeared in the DP deep learning literature. We anticipate (and very excited) that it is this aspect of private learning that will have a huge impact in practice. Indeed, a model trained using our framework is saving millions of $ of inference cost in a large organization as we wrote in Lines [85, 92]. It is a message worth spreading. Given a venue like ICLR, we anticipate that this work will be a harbinger of a new direction in privacy preserving deep learning.

---

> > ### Author Response · Authors · 2023-11-21
> > **Rebuttal Part 2: Answers to specific questions**
> >
> > Q: The technical innovation in this study is relatively constrained. The idea of training a classifier for selecting better data has been adopted in GPT3. Previous works, such as [1], have already introduced domain-specific pretraining, and techniques for data selection in training language models have been introduced in [2, 3]. The innovation of this work is the private learning of the domain classifier, which is limited.
> >
> > There are two important differences compared to previous literature.
> >
> > Use of a classifier to do data selection has been explored in other scenarios such as GPT3 paper; we indeed cite this in the beginning of Section 3.1. However, there is a major conceptual difference. In the GPT3 paper it was primarily used as a data clearing routine. Our focus is on how data selection changes the model size vs performance.  We think that this is a fundamental observation, and relates to quality of tokens and downstream performance and its effect on private fine-tuning.
> >
> > Second, we show that selective pretraining is more important for private deep learning compared to non-private deep learning [Figure 4]. This is an important message that is not known in the private deep learning literature.
> >
> > Similar ideas appear in  (a) “Intelligent Selection of Language Model Training Data” and (b)“Don’t Stop Pretraining: Adapt Language Models to Domains and Tasks” to our work.
> >
> > There are some similarities between these works and ours and we will consider citing them in the future revisions. We have read these papers. The paper (a) is not concerned with deep learning models but studies n-gram models. On the other hand, (b) studies how one can improve the performance of LLMs such as RoBERTA for a target task. Here they propose two techniques for continuing pretraining: 1) domain adaptive pretraining (DAPT) and 2) task adaptive pretraining (TAPT), which loosely speaking say that continuing pretraining on the target data helps.
> >
> > We see many differences between our work and these papers:
> >
> > We are proposing how to use task specific data to do a better way of pretraining  from scratch ( = a randomly initialized model).   This is not the same as continuing the pretraining. In fact, both DAPT and TAPT start with a fully pretrained RoBERTA. On the other hand, we do not even train on all the tokens in the pretraining data but only a small subset.
> > Our main focus is on how pretraining data quality affects the model size. In particular, our hypothesis is that when the models are small, pretraining data needs to be better and have fewer tokens. Both (a) and (b) do not study this aspect.
> > Finally, we are in the world of private learning while those papers are not.
> >
> > We believe that a fairer comparison (in the non-private world) is the concurrent work of Xie et al [77]. However, they do not study model size tradeoffs with data selection, which is our emphasis.

---

> > > ### Author Response · Authors · 2023-11-21
> > > **Rebuttal Part 3: Missing baselines,  DP calibration**
> > >
> > > Q: Some other baseline are missing: if one knows the domain of the private task, then one could select pretraining data in that domain without having to perform the DP data selection.
> > >
> > > It is hard to find off-the-shelf pre-training data for most of the real-word private datasets. For example, for real world emails, there are no large-scale opensource email corpus. Therefore, we argue that performing DP data selection is necessary for those private datasets. Nonetheless, if the reviewer has some concrete suggestions, could you please explain further, and we would love to discuss.
> > >
> > >
> > > Q: zero-shot learning of compressed or quantized larger models.
> > >
> > > We compare against the zero-shot learning of larger models. For example, consider Figure 1, where we compare the zero-shot performance of GPT2.  Note that GPT2 is a much stronger baseline than a compressed GPT2 or quantized GPT2. So, we think that we have covered the baselines you are asking. For GLUE, in Figure 5, we compare against DistillBERT, which is again a strong baseline.  We hope this clarifies your question.
> > >
> > > We do not explicitly compare against quantized models. However, quantization can only hurt the performance. Hence, our zero-shot comparison of GPT2 is a stronger baseline.
> > >
> > > Q: One can also select the public dataset with DP to pretrain a large model, and then compress and finetune it on private data. How does this compare with pretraining a smaller model?
> > >
> > > First, if we can use a larger model, then selective pretraining is less important. Main message of the paper is that when model size is small, the quality of pretraining tokens matters more, and hence we need selective pretraining. Having said that, even if we assume we have trained a large model, how to compress it to a small model is unclear. For example, previous works such as Mireshghallah et  al have explored DP model compression strategies and concluded that out of the box compression algorithms that work very well in the non-private world do not extend to DP world.
> > >
> > > It is precisely the fact that we do not know how to compress a larger model to a smaller model in the DP world, we believe that our contribution is significant. We give a novel framework on how to train small models preserving DP. This is also the reason we titled our paper as model compression, because in spirit we show how to train a small model.
> > > We note that the best performing small model in our experiments, DistilBERT, is a distilled version of BERT model using KD algorithm, which is same as pretraining. We show a better way of pretraining for private finetuning that beats DistillBERT. It is for this reason we call our technique a compression technique.
> > >
> > > Q: DP training is known for introducing worse calibration to the classifier. How might that impact the data selection process?
> > >
> > > Although deep neural networks could be overconfident and need calibration in some applications (Guo et al., 2017; Zhang et al., 2022), not calibrating the outputs *does not* affect our algorithm. This is because we rank all sequences according to the confidence scores, and calibration does not change the relative ranking. We discussed this in the first paragraph of Section 3.1 in our original submission.
> > >
> > > If our answers change your evaluation of the paper, please consider raising the score.
> > > We believe that our paper has an important message for the DP + LLM community, and venues like ICLR are an appropriate place to convey this.

---

### Meta-Review · Area_Chair_jqEK · 2023-12-05

**Metareview:**

This paper studies the question how (selective) pre-training (on a public dataset) can be used to improve the privacy-utility-efficiency trade-off of differentially private fine-tuning of text prediction models (on a private dataset). More precisely, the paper considers a 3-step recipe consisting of privacy-preserving data selection, followed by non-private pre-training, and then by private fine-tuning (using the DP-SGD algorithm of Abadi et al.). The privacy-preserving data selection step is carried out by initializing a classifier with a pre-trained language model and fine-tuning it with differential privacy on a dataset where training examples from the private data are positively labeled, and training examples sampled from the other source of data are negatively labeled.

The paper focuses on compressed models which are cheaper to serve, and demonstrates that using part of the privacy budget to select which training examples from the public dataset to train on and which to skip can yield a superior privacy-utility trade-off compared to pre-training on the full dataset or on a random subset of it.

The problem studied in this paper is interesting and timely, and the paper is well-written.
As pointed out by one of the reviewers, the idea of selective pre-training was explored before in the non-private setting, and is not restricted to the private setting. The fact that it works well in the private setting is interesting and worth being published. However, it falls short of the novelty bar for the ICLR.

**Justification For Why Not Higher Score:**

The problem studied is interesting, but the novelty of the paper is limited.

**Justification For Why Not Lower Score:**

n/a

---

### Decision · Program_Chairs · 2024-01-16

Reject